# Retrograde and Anterograde Transport of Lat-Vesicles during the Immunological Synapse Formation: Defining the Finely-Tuned Mechanism

**DOI:** 10.3390/cells10020359

**Published:** 2021-02-09

**Authors:** Juan José Saez, Stephanie Dogniaux, Massiullah Shafaq-Zadah, Ludger Johannes, Claire Hivroz, Andrés Ernesto Zucchetti

**Affiliations:** 1Institut Curie, Université PSL, U932 INSERM, Integrative Analysis of T Cell Activation Team, 26 Rue d’Ulm, 75248 Paris CEDEX 05, France; juan-jose.saez-pons@curie.fr (J.J.S.); Stephanie.Dogniaux@curie.fr (S.D.); 2Institut Curie, Université PSL, U1143 INSERM, UMR3666 CNRS, Cellular and Chemical Biology Unit, Endocytic Trafficking and Intracellular Delivery Team, 75005 Paris, France; massiullah.shafaq-zadah@curie.fr (M.S.-Z.); Ludger.Johannes@curie.fr (L.J.)

**Keywords:** LAT, T-cell activation, immune synapse

## Abstract

LAT is an important player of the signaling cascade induced by TCR activation. This adapter molecule is present at the plasma membrane of T lymphocytes and more abundantly in intracellular compartments. Upon T cell activation the intracellular pool of LAT is recruited to the immune synapse (IS). We previously described two pathways controlling LAT trafficking: retrograde transport from endosomes to the TGN, and anterograde traffic from the Golgi to the IS. We address the specific role of four proteins, the GTPase Rab6, the t-SNARE syntaxin-16, the v-SNARE VAMP7 and the golgin GMAP210, in each pathway. Using different methods (endocytosis and Golgi trap assays, confocal and TIRF microscopy, TCR-signalosome pull down) we show that syntaxin-16 is regulating the retrograde transport of LAT whereas VAMP7 is regulating the anterograde transport. Moreover, GMAP210 and Rab6, known to contribute to both pathways, are in our cellular context, specifically and respectively, involved in anterograde and retrograde transport of LAT. Altogether, our data describe how retrograde and anterograde pathways coordinate LAT enrichment at the IS and point to the Golgi as a central hub for the polarized recruitment of LAT to the IS. The role that this finely-tuned transport of signaling molecules plays in T-cell activation is discussed.

## 1. Introduction

T cell activation is key in establishing an adaptive immune response and it requires a direct interaction between T cells and antigen presenting cells (APC), where the T cell receptor (TCR) recognizes a peptide presented by MHC molecules on the APC [1,2]. The contact region between both cells forms the immune synapse (IS), where both endocytic and exocytic vesicular trafficking focus [3]. This regulated transport plays a key role in T cell activation since it controls the polarized secretion and concentration towards the interacting cells of cytokines [4,5], extracellular vesicles [6,7,8] and receptors and ligands such as CD40L [9,10], thus shaping the adaptive immune response. The endocytic and exocytic traffic of vesicular compartments has also been shown to regulate TCR-induced signaling [11]. Although the engagement of the TCR takes place at the plasma membrane, the TCR/CD3 complex and its associated signaling machinery are also present in intracellular vesicles. Different studies, including ours, have shown that molecules involved in T cell signaling are present both at the plasma membrane and in intracellular vesicular pools [12,13,14]. Remarkably, even though the signaling mediators are recruited to the IS, they follow different endocytic and exocytic routes. Deciphering the nature of these trafficking pathways and the molecular machinery involved is important for understanding the role(s) of this traffic in T cell activation.

Our team analyzed the traffic to the immune synapse of the linker for activation of T cells (LAT). LAT is a transmembrane protein that plays a key role in T lymphocyte signaling and function [15,16,17,18]. This adaptor protein is phosphorylated on multiple tyrosines upon TCR activation and forms complexes with numerous proteins, also known as LAT signalosomes [19]. It is present at the plasma membrane and more abundantly on intracellular membranes [11,12,13]. The purification of LAT-containing vesicles performed by our team [20] and their proteomic analysis revealed the presence of several molecules involved in membrane trafficking of cargo proteins. Indeed, we previously showed that the vesicular SNARE VAMP7 [13], the small GTPase Rab6 [21], the target SNARE syntaxin-16 (Synt16) [21] and the Golgin GMAP210 [22] were present on LAT vesicles and that they all controlled the recruitment of LAT to the immune synapse as well as some aspects of the TCR-induced activation of T lymphocytes. One of our studies pointed to the role of Rab6 and Synt16 in the retrograde transport of LAT from endosomes to the Golgi [21]. Two other studies suggested that VAMP7 and GMAP210 were involved in the anterograde transport of LAT to the immune synapse [13,22]. Although these findings highlighted the importance of the spatial organization of vesicular trafficking in the recruitment of LAT to the immune synapse and described molecules and pathways involved in this traffic, some “pieces of the puzzle” were missing. In particular some molecules, i.e., Rab6 and GMAP210 have been shown to play a role in both retrograde and anterograde transport of cargoes [23,24,25,26,27,28] and we wanted to clarify their involvement in LAT trafficking.

In the present study we show that in activated T lymphocytes Rab6 is specifically involved and restricted to the retrograde transport of LAT, whereas GMAP210 is specifically involved in the anterograde Golgi to IS delivery of LAT. Altogether our data clarify the finely tuned mechanisms of LAT-trafficking.

## 2. Materials and Methods

### 2.1. Cells

Jurkat T cells (validated by SSTR method present 88% of homology with Jurkat ACC 282 with DSMZ website), and Raji B (ATCC, CCL-86) cells were cultured at 37 °C 5% CO_2_ in RPMI 1640 Glutamax (61870-010, Gibco, Life Technologies, Paisley, UK) supplemented with 10% fetal calf serum (FCS, CVFSVF00-01, Eurobio, Paris, France) and were passed every 2–3 days at ~0.5 × 10^6^ cells/mL.

### 2.2. Reagents and Antibodies

Recombinant Staphyloccocus Enterotoxin type E (SEE, Cellgenetech, MBS1112600), Poly-L-lysine (Poly-Lys, P8920, Sigma-Aldrich, St. Louis, MO, USA) and Mitotracker (M22426, Invitrogen, Grand Island, NY, USA) were used.

For immunofluorescence we used the following primary antibodies: rabbit anti-LAT (5 µg/mL, 06–807; Millipore), human anti-GFP (produced by the Institut Curie platform), mouse anti-VAMP7 (1:400, NBP1-07118, Novus Biologicals, Centennial, CO, USA), rabbit anti-Rab6 (1:400, 9625; Cell Signaling Danvers, MA, USA), rabbit anti-Synt16 (5 µg/mL, ab134945; Abcam, Cambridge, UK) or human anti-Giantin (produced by the Institut Curie platform). As secondary antibodies we used (1:300) anti-mouse Ig Alexa Fluor 568 (A11004; Thermo Fisher Scientific, Waltham, MA, USA), anti-human Ig Alexa Fluor 488 (A11013, Thermo Fisher Scientific, Waltham, MA, USA) or anti–rabbit Ig Alexa Fluor 568 (A11036; Thermo Fisher Scientific Waltham, MA, USA), according to primary antibody species.

For Immunoblot, we used the following primary antibodies: rat anti-α-tubulin (1:1000, AbC117-7285, AbCys, Abcam, Cambridge, MA, USA), rabbit anti-PLCγ1 (1:1000, 2822S, Cell Signaling, Danvers, MA, USA), rabbit anti-ZAP70 (1:1000, 3165S, Cell Signaling, Danvers, MA, USA), rabbit anti-LAT (5 µg/ml, 06–807; Millipore, Merck Darmstadt, Germmany), rabbit anti-CD28 (1:1000, 38774, Cell Signaling, Danvers, MA, USA), rabbit anti-GMAP210 (Gift from Michel Bornens), rabbit anti-VAMP7 (Gift from Thierry Galli), rabbit anti-Rab6 (0.2 µg/mL, sc-310; Santa Cruz Biotechnology, Dallas, TX, USA), rabbit anti-Synt16 (0.1 µg/mL, ab134945; Abcam, Cambridge, MA, USA), rat anti-gp96 (1:1000, ADI-SPA-850; Enzo, New York, NY, USA), rabbit anti-GM130 (1:1000, ab52649; Abcam, Cambridge, MA, USA), and mouse anti-Vps-35 (1:1000, ab10099; Abcam, Cambridge, MA, USA). As secondary antibodies we used (1:10,000; Jackson ImmunoResearch, West Grove, PA, USA): anti–rabbit HRP (111-036-046) or anti–rat HRP (112-035-143) according to primary antibody species.

### 2.3. Production of Lentiviruses and Infection of Jurkat Cells

Nonreplicative VSV-g pseudotyped lentiviral particles were produced by transfecting HEK-293T cells with Gag, Pol, rev, encoding plasmid (pPAX2), envelop encoding plasmid (pMD2.G) and either the HA-Tev-LAT construct [13] encoded in a pWXLD vector, or the SNAP-GalT-GFP construct [29] encoded in a pCDH-EF1-MCS-IRES-Puro vector (SBI System Biosciences, Embarcadero Way, Palo Alto, CA, USA), or short hairpin RNA (shRNA) sequences encoded in pLKO.1 plasmid: Nontargeting control shRNA (shC, Mission shRNA SHC002, Sigma-Aldrich, St. Louis, MO, USA), GMAP210 shRNA (Mission shRNA, TRCN0000022021, Sigma-Aldrich, St. Louis, MO, USA), VAMP7 shRNA (Mission shRNA, TRCN0000059892, Sigma-Aldrich, St. Louis, MO, USA), Rab6 shRNA (Mission shRNA, TRCN0000379588, Sigma-Aldrich, St. Louis, MO, USA), and Synt16 shRNA (Mission shRNA, TRCN0000161930, Sigma-Aldrich, St. Louis, MO, USA). Lentivirus were recovered in supernatant after 2 days and concentrated. 5 × 10^6^ Jurkat T cells were infected for 24 h, cells infected with shRNA encoding virus were selected in puromycin (2 µg/mL, ant-pr, InvivoGen, Toulouse France) and used 4 days postinfection.

### 2.4. Plasmids and Transfection

The plasmid encoding HA-TeV-LAT, LAT-mCherry and VAMP7-GFP were reported previously [13,22]. The plasmids encoding chimeric molecules GMAP210-GFP, GFP-GMAP-Mit and GFP-Mit are described elsewhere [30]. The plasmid encoding Rab6A-GFP was a gift from Bruno Goud (Institut Curie, CNRS-UMR144, Paris, France). Jurkat T cells were transfected using the Amaxa Cell Line Nucleoefector Kit V (VCA-1003, Lonza, Basel, Switzerland). Transient transfections were performed as follows, 5 × 10^6^ cells were washed, resuspended in 100 µL of nucleoefector solution and combined with 5–10 µg of DNA. Cells were passed into the electroporation cuvettes and electroporated (Amaxa program X-005), then incubated at room temperature for 10 min, recovered and diluted in warmed RPMI supplemented with 10% FCS and cultured for 24 h at 37 °C, 5% CO_2_.

### 2.5. Preparation of Lysates from Jurkat T Cells

A sample of 1 × 10^6^ cells/mL of Jurkat T cells was washed three times with cold PBS and incubated on ice for 20 min in 30 µL ice-cold lysis buffer (50 mM Tris pH 8, 150 mM NaCl, 1,5 mM MgCl2, 1% Glycerol, 1% TritonX100, 0.5 mM EDTA pH 8, 5 mM NaF) supplemented with a protease inhibitor cocktail (11873580001, Sigma-Aldrich, St. Louis, MO, USA). Postnuclear lysates were obtained by centrifugation at maximum velocity for 15 min at 4 °C. Laemmli Sample Buffer (161-0747, BIORAD, Hercules, CA, USA) and reducing agent (NP0009, Thermo Fisher Scientific Waltham, MA, USA) were added and samples were heated at 95 °C for 5 min and kept at −20 °C before immunoblot analysis.

### 2.6. Purification of LAT-Signalosome

Jurkat cells (5 × 10^6^) were resuspended in 200 µL of RPMI medium, and magnetic beads (1 × 10^7^) coated with anti-CD3 and anti-CD28 (11132D, Gibco, Thermo Fisher, Waltham, MA, USA) were added in a volume of 100 µL. Beads were incubated with T cells for the appropriate time at 37 °C. Activation was stopped with the addition of 500 µL cold PBS, and 80 µL (1/10) of samples were collected as “input” and lysed as described above. Bead-cell conjugates were then magnetically restrained, resuspended in 500 µL of “freeze–thaw” buffer (600 mM KCl, 20 mM Tris, pH 7.4, and 20% glycerol) supplemented with EDTA-free Protease Inhibitor Cocktail Tablet (1123000, Roche, Mannheim, Germanny). Samples were submitted to seven cycles of freezing and thawing. After the final cycle, 5 µL benzonase (71206-3, Millipore, Merck Darmstadt, Germmany) was added, followed by incubation for 20 min at room temperature. Samples were magnetically restrained to purify the bead-attached membranes and then were washed five times in the supplemented “freeze–thaw” buffer described above. Bead-associated proteins were resuspended in lysis buffer and separated by SDS–PAGE and analyzed by immunoblot.

### 2.7. Immunoblot Analysis

Samples were resolved on Mini-PROTEAN TGX Stain-Free Gels 4–15% (#4568086, BioRad, Hercules, CA, USA) and semi-dry transferred (TransBlot Turbo Transfer System, BioRad, Hercules, CA, USA) on PVDF membranes (162-0177 BioRad, Hercules, CA, USA). After blocking with TBS, 0.05% Tween20, 5% BSA for 1h on a rocking platform shaker, membranes were incubated overnight at 4 °C with primary antibodies. Membranes were washed three times with TBS 0.05% Tween20 and incubated for 1h in TBS, 0.05% Tween20, 5% BSA on a rocking platform shaker with the secondary antibody. Bound antibodies were revealed using the Clarity Western ECL substrate (#170-5061, BioRad, Hercules, CA, USA), according to the manufacturers’ directions. The intensity of the bands was quantified by signal intensity in ImageJ.

### 2.8. Coverslips and Dishes Preparation for Immunofluorescence Assay

Coverslips of 12mm ø (VWR, 631-0666) for fixed cells or fluorodishes (World Precision Instrument Inc., Sarasota, FL, USA, FD35-100) for live imaging were precoated with 0.02% of poly-L- Lysine for 20 min at room temperature and were washed three times in water before being dried and kept for a maximum of 2 days.

### 2.9. Preparation of Jurkat T Cells and Raji B Cells Conjugates

Raji B cells were washed, resuspended at a concentration of 1 × 10^6^ cells/mL in RPMI without FCS and labeled with CellTracker Blue CMAC dye (10 µM, C2110, Thermo Fisher Scientific, Waltham, MA, USA) for 20 min at 37 °C. Labeling was stopped with RPMI 10% FCS and cells were washed once and resuspended at 1 × 10^6^ cells/mL. Cells were pulsed with SEE (100 ng/mL) or left untreated for 30 min at 37 °C before being washed once and resuspended at a concentration of 1 × 10^6^ cells/mL. Some 100,000 Raji cells were incubated on coverslips for 30 min, washed once with warmed PBS and 150,000 Jurkat cells resuspended in RPMI 10% FCS were added for 30 min. Coverslips were washed once with cold PBS before fixation.

### 2.10. SNAP-Tag Capture Assay

Jurkat cells expressing both the GalT-GFP-SNAP and the HA-LAT encoding constructs were incubated for 30 min at 4 °C on a wheel with 1:100 of mouse anti-HA Ab (901515; Biolegend, San Diego, CA, USA). The cells were washed with cold PBS and incubated for 45 min at 4 °C on a wheel with membrane-impermeable BG–PEG9–NHS [29]. Cells were then incubated 4 h at 37 °C in complete medium to analyze constitutive traffic of endocytosed LAT or put on slides together with Raji B cells for 30 min to form conjugates as described in the “Preparation of Jurkat T cells and Raji B cells conjugates” section. Cells were then fixed and permeabilized and stained with anti-GFP (human anti-GFP from Institut Curie) to reveal the GalT-GFP-SNAP and anti-mouse Ig to reveal the anti-HA Ab. Colocalization of the proteins was analyzed as described in the “Analysis of HA-LAT trapping in Golgi in cells expressing SNAP-Tag” section.

### 2.11. Mitochondrial Capture Assay in Cells Expressing GFP-GMAP-Mit

Jurkat cells were washed, resuspended at 1 × 10^6^ cells/mL and incubated 4 h with nocodazol (5 μg/mL, M1404, Sigma-Aldrich, St. Louis, MO, USA) in RPMI containing 10% of FCS at 37 °C. 150,000 Jurkat cells were then incubated on coverslip for 30 min, washed once with cold PBS and fixed.

### 2.12. Live TIRF Microscopy

After coating with poly-L-Lysine, fluorodishes were coated overnight at 4 °C with αCD3ε (OKT3, #16-0037-85, eBioscience, Thermo Fisher Scientific, Waltham, MA, USA) + αCD28 (CD28.2, #302914 Biolegend, San Diego, CA, USA), washed three times and prewarmed at 37 °C for 10–15 min. 200,000 Jurkat T cells were plated and images for 491 and 561 channels were acquired every 5 s.

### 2.13. Fixation

Cells were fixed with 4% paraformaldehyde (FB002, Life technologies, Grand Island, NY, USA) for 10 min at room temperature, washed once in PBS and excess of paraformaldehyde was quenched for 10 min with PBS 10 mM Glycine (G8898, Thermo Fisher Scientific, Waltham, MA, USA). Coverslips were kept at 4 °C in PBS until permeabilization and staining.

### 2.14. Staining and Mounting

Cells were permeabilized for 30 min at room temperature with PBS 0.2% Bovine Serum Albumin (BSA, 04-100-812Euromedex, Souffelweyersheim, France) and 0.05% Saponin (S4521, Sigma-Aldrich, St. Louis, MO, USA). Cells were then incubated for 1 h at room temperature with primary antibody, followed by washing three times with PBS 0.2% BSA 0.05% Saponin and incubated protected from light for 30 min in the same buffer with spun secondary antibodies. After washing once with PBS BSA Saponin, and once with PBS, coverslips were soaked three times in PBS, three times in water, and mounted on slides. For VAMP7 endogenous staining, cells were treated as in Larghi et al. (2013). For regular confocal microscopy, coverslips were mounted with 4–6 µL of Fluoromount G (0100-01, SouthernBiotech, Birmingham, AL, USA) on slides (KNITTEL Starfrost) and dried overnight protected from light before microscope acquisition.

### 2.15. Microscopes and Images Analysis

Images were acquired using an inverted laser scanning confocal microscope (Leica DMi8, SP8 scanning head unit), equipped with HC PL APO CS2 63x/1.40 OIL objective. TIRF microscopy was performed using an inverted Nikon microscope Ti-E from the Nikon Imaging Center at Institut Curie-CNRS equipped with a 100X CFI Apo TIRF objective (numerical aperture of 1.49), 491 and 561 nm lasers, and an EMCCD 512 Evolve camera (Photometrics, pixel size 0.16 µm). For live experiments, the temperature was constantly sustained at 37 °C and one image was acquired every 5 s. Images were analyzed on Fiji and ImageJ software and compatible scripts were generated for automated or semi-automated analysis.

### 2.16. Recruitment at the Immune Synapse and “Mean Cell” Creation

Single images corresponding to the middle planes of conjugates were extracted from Z-stack. T cells were cropped and oriented in the same way regarding their synapse. Obtained T cell images were grouped by condition (shRNA ± SEE) and fluorescence intensities were normalized by the mean fluorescence intensity (MFI). Images were then resized to the smallest image size in order to create a normalized stack of images for each group. All groups were normalized (size and intensity) before being compared. Stacks of aligned cells were finally projected (averaging method) giving single plane “mean cells”. Stacks were resized to obtain a 1-pixel height stack by averaging the fluorescence intensity of the total height of each image. Projections of the 1-pixel resized stacks were obtained based on average and standard deviation methods and pixel intensity profiles were performed along projection’s width. In order to get a cell-by-cell quantification, we also computed an enrichment ratio at the synapse. This enrichment was defined as the ratio between the total cell fluorescence and the fluorescence in the synaptic region (rectangle at the synapse representing 20% of the total cell).

### 2.17. Analysis of Molecule Capture in the Mitochondria

Z-stack (0.5 µm) images of similarly dimensioned Jurkat cells were chosen. In that z-stack, cells were manually segmented to obtain a ROI. In each ROI, masks based on both GFP [11] and VAMP7/Rab6/Synt16 stainings were created by thresholding. Automatic colocalization assays were performed with Mander’s overlap coefficient, using the JACoP plugin for ImageJ64.

### 2.18. Analysis of HA-LAT Trapping in Golgi in Cells Expressing SNAP-Tag

Z-stack (0.5 µm) images of similarly dimensioned Jurkat cells were chosen. In that z-stack, a ROI surrounding the Golgi was defined based on GalT-GFP-SNAP staining. Within each ROI, masks based on both GalT-GFP-SNAP and HA-LAT stainings were created by thresholding. Automatic colocalization assays were performed with Mander’s overlap coefficient, using the JACoP plugin for ImageJ64. Representative images show the summed intensity from z-projections of three planes of the z-stack containing the Golgi apparatus.

### 2.19. Flow Cytometry

Cells were centrifuged and transferred to conical bottom plate (Greiner Bio-One, 650101), stained for 20 min in cold PBS with Fixable Violet Dead Cell Stain Kit (1/4000, Invitrogen, L34955) and washed in FACS Buffer (PBS 0.5% BSA 2 mM EDTA). Extracellular staining was performed in FACS buffer for 30 min on ice. After staining, cells were washed in FACS buffer and stained with the corresponding secondary antibody. Finally, cells and compensation beads (eBioscience, 01-1111-42) were acquired with MACS Quant (Miltenyi) flow cytometer and data were analyzed with FlowJo software.

### 2.20. Statistical Analysis

Statistical analysis was performed with GraphPad Prism 7 software. If the data represented a Gaussian distribution, we applied a two-way ANOVA followed by Sidak’s multiple comparison test. For non-Gaussian distributed data, we used the nonparametric Kruskal−Wallis test followed by Dunn’s multiple comparison test. Data were considered statistically significant if the *p*-value obtained was lower than 0.05.

## 3. Results

### 3.1. Endocytosed LAT Is Recruited to the Immune Synapse: Regulation by Retrograde and Anterograde Pathways

We have previously observed that LAT undergoes endocytosis from the plasma membrane to intracellular compartments from where it can be recruited to the immune synapse (IS) under TCR triggering [14,21]. To better characterize the recruitment of this endocytic pool of LAT, we studied the kinetic of its recruitment to the IS. To do so, we used Jurkat T cells expressing a chimeric LAT protein containing HA-tagged in its extracellular domain, described elsewhere [13]. We labeled the plasma membrane pool of HA-LAT with a mouse anti-HA antibody (Ab), washed out the unbound antibody from the cells, and allowed its endocytosis for 4h to stain the internalized pool of LAT (Figure 1A). Finally, cells were seeded on glass slides together with Raji B cells and the conjugates formed were activated with SEE for different time points and then fixed. As expected, the pool of LAT, which is internalized from the plasma membrane, is enriched in intracellular compartments. Upon activation, i.e., in the presence of TCR activation, the intracellular pool of LAT is recruited to the IS. This recruitment follows a dynamic similar to the enrichment of the total LAT at the IS in non-transduced Jurkat cells (representative images in Figure 1B). To confirm this result, we quantified total LAT and endocytic LAT enrichment at the immune synapse in Jurkat T cells interacting with Raji B and activated with SEE for different time points (average density map representation in Figure 1C and quantification in Figure 1D). Of note the enrichment of total LAT and endocytic LAT at the IS were very similar. This correlates with the abundance of this endocytic pool, i.e., around 70% of the total LAT is intracellular [11]. Moreover, in both cases the distribution of LAT (central or peripheral) was also very similar. We next tested whether the recruitment of the endocytic pool of LAT depended on the retrograde and anterograde pathways. In Jurkat T cells expressing HA-LAT, we silenced GMAP210, VAMP7, Rab6 and Synt16, which were described by us to regulate anterograde and retrograde traffic of LAT during IS formation [13,21,22]. Using lentivirus encoding different short hairpin RNAs (shRNAs), we achieved a reduction of protein expression to less than 40% of the control group (Appendix A). We first evaluated the effect of GMAP210, VAMP7, Rab6 or Synt16-silencing on LAT endocytosis. To do so, the surface pool of LAT was labeled in HA-LAT-Jurkat cells with anti-HA Ab and we imaged the cells before and after 4h incubation at 37 °C to allow LAT internalization. FACS analysis showed that all silencing conditions presented a similar level of HA-LAT protein expression at the plasma membrane (Appendix A). Moreover, GMAP210, VAMP7, Rab6 and Synt16-silencing did not block the endocytosis of LAT (Figure 2A, right panel). We then tested the dynamic of the internalized pool of LAT during immune synapse formation. We took HA-LAT expressing Jurkat cells which were extracellularly labeled and incubated for 4h at 37 °C, to stain the endocytic pool of LAT, and activated them with SEE-pulsed Raji cells. In all the conditions of silencing, we observed an impairment of the recruitment of the endocytic pool of LAT to the IS after activation (representative images in Figure 2B). This was consistent with our previous work on total LAT recruitment. To confirm this result, we quantified HA-LAT enrichment at the immune synapse in Jurkat T cells interacting with Raji B cells in the absence (no synapse formation) or presence of SEE (formation of the immune synapse). In this model, recruitment of the endocytic pool of LAT was effectively decreased when GMAP210, VAMP7, Rab6 or Synt16 were silenced (average density map representation Figure 2C and quantification in Figure 2D). Of note although LAT enrichment, to the IS, was lower in cells expressing the ShGMAP210 than in control cells, it was still induced by SEE (Appendix A). This was not the case in cells expressing ShRNAs specific for VAMP7, Rab6 or Synt16 (Appendix A). This might be due to the difference in silencing efficiencies of ShRNAs, the ShRNA targeting GMAP210 being less efficient (Appendix A). Altogether, these results demonstrate that the endocytic pool of LAT is recruited to the IS during activation and this process specifically relies on GMAP210 and VAMP7 regulators when it concerns the anterograde pathway, and Rab6 and Synt16 molecules when it implies the retrograde pathway.

### 3.2. GMAP210 and VAMP7 Are Recruited to the IS Together with LAT; Rab6 and Syntaxin 16 Are Not

We next addressed the relative role of each of the molecules in the recruitment of the endocytic pool of LAT. Indeed, in our previous studies we showed that Rab6 and Synt16 silencing blocked the retrograde traffic of LAT-vesicles from the plasma membrane to the Golgi-TGN, but we did not address if they directly affected the anterograde transport of LAT. For example Rab6 has also been shown in other cell types to regulate the anterograde secretory pathway [24,27]. On the other hand, in activated T lymphocytes, our results suggested that GMAP210 and VAMP7 control the anterograde traffic of LAT from the Golgi to the IS. Yet, we did not address if they could also be involved in the retrograde transport of endocytic LAT since the two pathways control and assure the polarized delivery of LAT at the IS [13,21,22]. To better characterize the role of these molecules in each pathway and their association with LAT-vesicles during IS formation, we dynamically imaged the recruitment of LAT together with GMAP210, VAMP7 or Rab6 to the activation sites by TIRF microscopy. Jurkat cells were transfected to express Lat-mCherry together with GMAP210-GFP, VAMP7-GFP or Rab6A-GFP, placed on glass slides coated with activating anti-CD3/anti-CD28 antibodies and imaged when cells spread. The dynamic analysis of the recruitment of LAT and GMAP210, VAMP7 or Rab6 to the activation sites showed that the pool of LAT polarized towards the synapse was close to the compartments labeled with GMAP210, VAMP7 and Rab6A (Figure 3A and Appendix A). Moreover, activation induced the appearance of new LAT-vesicles moving toward the periphery and backward. LAT arrival in the evanescent field coincided with GMAP210 and VAMP7 arrival and LAT spots also moved together with GMAP210 and VAMP7 spots suggesting that vesicles containing LAT also contained GMAP210 and VAMP7 (Figure 3A still images, Appendix A). In contrast, only discrete colocalization was observed between LAT-vesicles and Rab6 and the two labeling “did not move together” from the center to the periphery suggesting that the pool of LAT recruited to the immune synapse did not contain Rab6 (Figure 3A still images, Appendix A).

To further confirm these results, we biochemically analyzed the recruitment of the different molecules in the activation sites. To do so, we activated Jurkat T cells with magnetic beads coated with anti-CD3/anti-CD28 antibodies for different times, broke the cells with repeated cycles of freezing and thawing and purified the membranes attached to the beads with a magnet. Western blot analysis of the bead-associated complexes revealed the presence of LAT, as well as different signaling molecules that have been shown to be part of the LAT signalosome: the tyrosine kinase ZAP70, the phospholipase PLCγ1 [13]. It also showed, as control, the presence of CD3ε and CD28, which are “precipitated” with the beads coated with anti-CD3ε and anti-CD28 antibodies (Figure 3B). As shown previously [13,22], it also revealed the recruitment of both GMAP210 and VAMP7 with a kinetic similar to LAT recruitment. In contrast, we did not detect Rab6, Synt16 and Vps35, a major component of the retromer core complex and notably involved in the retrograde transport of cargoes [28]. These results confirmed the microscopy analysis obtained for Rab6 (Appendix A and Figure 2A).

Altogether, these results show that GMAP210 and VAMP7, regulators of the anterograde pathway, are recruited on vesicles containing LAT and are involved in LAT recruitment to the immune synapse. In contrast, Rab6 and Synt16, are not physically associated to the LAT vesicles and are not recruited to the IS, although they also regulate LAT recruitment (Figure 2C,D).

### 3.3. GMAP210 and VAMP7 Are Not Involved in the Regulation of the Retrograde Pathway

We then wanted to test if GMAP210 and VAMP7 can be involved in the retrograde transport of LAT from endosome to the TGN-Golgi. To do so, we used the capture assay we had previously developed [21,29], which allows retention and concentration in the Golgi of the pool of LAT following this pathway. Briefly, Jurkat T cells expressing the HA-LAT, were silenced for GMAP210, VAMP7, Rab6 or Synt16 and then infected with a lentiviral vector encoding for the Golgi resident galactosyl transferase (GalT) protein fused with the GFP-SNAP capture reagent (GalT-GFP-SNAP). After labeling of the cell surface pool of LAT at 4 °C with the anti-HA Ab and washing, the extracellularly exposed proteins were coupled with the non-membrane-permeable amino-reactive reagent BG-PEG_9_-NHS at 4 °C. After a second wash, cells were incubated with unpulsed or SEE-pulsed Raji B cells for 30 min at 37 °C. If transported to the Golgi via the retrograde route, BG-coupled proteins formed a covalent bond with the SNAP tag of the GalT-GFP-SNAP fusion protein and were thus captured in the Golgi (Figure 4A). To measure this capture, and thus the retrograde transport of endocytic HA-LAT, we quantitatively analyzed the colocalization of HA-LAT with GalT-GFP-SNAP on confocal images of fixed cells. As described [21], in resting conditions, i.e., in the absence of SEE, HA-LAT was not captured in the Golgi after 30 min of incubation at 37 °C (Appendix A). In activating conditions, i.e., in the presence of SEE, an accumulation of HA-LAT was observed in the Golgi of Jurkat T cells expressing control, GMAP210 and VAMP7 shRNAs. In contrast, silencing of both Rab6 and Synt16 significantly decreased the capture of the endocytosed pool of HA-LAT in the Golgi (representative images in Figure 4B and quantification in Figure 4C). These results suggest that GMAP210 and VAMP7 do not participate in the retrograde transport of LAT to the Golgi induced by TCR triggering.

In our previous work, we demonstrated that GMAP210 is able to interact and capture VAMP7-containing vesicles [22]. To further confirm that GMAP210 is not involved in the retrograde pathway, we tested if GMAP210 could capture vesicles containing Rab6 and Synt16 that were shown here to control the retrograde transport of LAT. To do so, we adopted a strategy developed by others [30,31] and used in [22]. Briefly, we artificially attached GMAP210 to mitochondria and measured the capture, by GMAP210, of different cargoes on these organelles (Figure 4D). Jurkat cells were transfected with a construct encoding a GFP chimeric GMAP210 molecule containing the C-terminal hydrophobic domain of ActA, which anchors GMAP210 to mitochondria. The GFP protein containing the same domain of ActA (referred as GFP-Mit) [30] was used as control. It was previously shown that depolymerization of microtubules favored the proximity between mitochondria and vesicles hence allowing the potential capture of vesicles by golgins present on mitochondria [31,32]. Transfected Jurkat cells were thus treated with nocodazole. As previously described [22], the ectopic localization of GMAP210 to mitochondria induced the displacement of VAMP7 to mitochondria. In contrast, Rab6 and Synt16 were not trapped by GMAP210 in the mitochondria (representative images Figure 4E and quantified in Figure 4F). Of note, the presence of GFP in the mitochondria did not induce any displacement of the analyzed molecules (Appendix A). These results strongly support that GMAP210 binds VAMP7 “decorated” vesicles and does not bind membranes containing proteins involved in the retrograde pathway such as Rab6 and Synt16. Altogether these results show that the retrograde and anterograde traffic of LAT are two mechanistically independent pathways supported by their own molecular machinery.

## 4. Discussion

LAT is localized at the plasma membrane and in intracellular vesicles in T lymphocytes [12]. Previous works demonstrated that the vesicular pool of LAT, which is more abundant than the plasma membrane pool, plays a critical role in T cell activation and formation of the LAT signalosome [11,13,33,34]. Whereas others showed that the pre-existing plasma membrane pool of LAT is recruited rapidly, phosphorylated at sites of T-cell activation and able to form a complete signalosome while the vesicular pool is recruited later and contributes to the maintenance of sustaining signaling [35]. Thus the relative contribution of each pool to LAT-dependent signaling and T cell activation is still a matter of debate.

Examining the intracellular traffic of the internal pool of LAT and its origin, we recently showed that LAT followed the retrograde trafficking route from the plasma membrane to the Golgi-TGN and that this retrograde transport was controlling LAT signalosome formation and T cell activation [21]. We also showed that recruitment of the intracellular pool of LAT to the immune synapse required the VAMP7 v-SNARE protein as well as the GMAP210 Golgin [13,22]. We herein addressed the fine tuning of this transport to get some insights into the relative contribution of each of the intracellular trafficking routes to the recruitment of the intracellular pool of LAT to the immune synapse and the formation of the LAT signalosome.

Using real-time imaging to track the internalized pool of LAT, we showed that this pool was rapidly recruited to the immune synapse upon TCR activation. Importantly, this recruitment depended on key membrane trafficking factors such as Synt16 and Rab6, involved in the retrograde transport of cargo proteins from the plasma membrane to the TGN-Golgi, as well as VAMP7 and GMAP210 implicated in the post-Golgi to plasma membrane anterograde trafficking (Figure 2).This was particularly intriguing for GMAP210 as this Golgin has been shown to tether vesicles at the cis-Golgi side and has been involved in the early biosynthetic-secretory pathway between the endoplasmic reticulum and the Golgi [25,36,37,38]. Our previous work [22] together with the results reported herein confirm an original function of GMAP210 that directly regulates the trafficking of vesicles in a post-Golgi process, ultimately controlling the immune synapse formation. This has also been shown for the transport of proteins to the primary cilium, which presents similarities with the immune synapse [39,40], likely indicating a cell-type specialization of this Golgin. Moreover, using a mitochondrial capture assay, we confirmed that GMAP210 bound vesicles bearing VAMP7 and show herein that it did not bind vesicles containing Rab6 or Synt16 (Figure 4B). Knowing that GMAP210 has a binding preference for highly curved liposomes (radius < 50 nm) with monounsaturated lipids [41], these results suggest that the VAMP7/LAT containing vesicles, which are recruited via GMAP210 to the immune synapse, may modify the lipidic landscape of the synapse by bringing lipids that are rare at the plasma membrane. Our results also suggest that the size and the lipidic composition of the LAT-vesicles coming from the retrograde pathway are different from those delivered at the immune synapse by the anterograde pathway since they do not bind GMAP210.

In this study, we wanted to better characterize the role of molecules that we previously described as regulators of LAT recruitment to the immune synapse. This was particularly important for the small Rab GTPase Rab6, which has been shown to control both retrograde and anterograde transport of cargoes [23,24,42,43,44]. Yet, TIRF video-microscopy did not show any association of Rab6 and LAT at the activation sites (Figure 3A and Appendix A). Moreover, precipitation of membranes associated with activating beads revealed the recruitment of LAT, together with VAMP7 and GMAP210 but did not show any recruitment of Rab6 (Figure 3B). These data suggest that Rab6 is only involved in the retrograde transport of LAT.

Our data allow a better definition and comprehension of the temporal sequence of events involved in LAT recruitment to the immune synapse. Once internalized, LAT is transported to the Golgi via the Synt16/Rab6 dependent retrograde pathway. It is then transported forward to the immune synapse in vesicles containing VAMP7, which are conveyed by the GMAP210 golgin.

We show herein that each pathway is controlled by its own set of proteins suggesting separated molecular mechanisms without any molecular machinery cross-talks. Indeed, our data suggest that neither GMAP210/VAMP7 is involved in the retrograde transport of LAT-vesicles to the Golgi apparatus, nor Rab6/Synt16 deliver LAT-vesicles to the IS in the anterograde pathway. Thus, our model suggests that under TCR-triggering, LAT is first endocytosed to endosomes from where it is transported to the Golgi apparatus in a Rab6/Synt16-dependent manner. Once in the Golgi, GMAP210 and VAMP7 direct the polarized delivery of LAT-vesicles to the IS (Figure 5).

The TCR/CD3 complex or at least some of the chains composing this complex are present in recycling endosomes, which are recruited to the IS upon activation [14,45]. LAT molecules are also present in recycling endosomes [12]. Yet, LAT and the TCR/CD3 complexes (at least some of the chains) present in the recycling endosomes follow different intracellular trafficking pathways to the IS. The endocytic pool of LAT first goes to the Golgi through the retrograde pathway before following a VAMP7-dependent polarized anterograde transport to the IS (Figure 5). This is not the case for the CD3ζ chain since, unlike LAT, CD3ζ recruitment to the IS is insensitive to the silencing of GMAP210 [22], Rab6 [21] and VAMP7 [13]. These results as well as previous studies showing that CD3ζ and LAT are present in different exocytic compartments [11] demonstrate that different signaling molecules involved in TCR triggering are not “traveling” together. They also suggest that they can form different signalosomes in different intracellular localizations [46].

However, we still do not know if the whole endocytic pool of LAT is following the pathway described in Figure 5. Indeed, recruitment of this pool to the IS is only partially inhibited by silencing of some of the targets studied here, i.e., GMAP210 (Figure 2C,D and Appendix A). This can either be due to the incomplete silencing of GMAP210 (Appendix A) or to the fact that part of the endocytic pool of LAT is directly polarized to the IS (like the TCR/CD3 complex). This would not be the first example. Indeed, the retrograde trafficking of the endocytic β1 integrin is restricted to the non-ligand-bound conformation of β1 integrin [47]. If the endocytic pool of LAT can follow different trafficking pathways, it will be important to find out what is/are the modification(s) (i.e., phosphorylation, palmitoylation) targeting LAT towards a given pathway.

This well controlled transport of LAT could ensure several functions in T cells [14]. It may avoid the lysosomal degradation of the internalized pool of LAT, which has been reported by others [48,49,50,51], using the Golgi compartment as a bioavailable reservoir. It may also induce the formation of a LAT-dependent signaling platform in the Golgi playing a specific role in T cell activation. Moreover, it may guarantee the re-palmitoylation of LAT, which has been shown by others and us to be essential for its cellular localization and scaffolding activity [51,52]. Of importance, the strategic positioning of the Golgi compartment, facing the IS, may also ensure the polarized resecretion of recycling LAT molecules to the IS after collecting them back in the Golgi apparatus, as shown for other cargoes [21,53,54]. By extension, this antero/retrograde transport loop might be shared by several nonidentified yet Lat-interacting partners forming the signalosome, making this trafficking route an intriguing and efficient pathway to properly organize the immunological synapse upon TCR activation.

Thus, our results suggest that LAT endocytosis and retrograde transport are major cellular processes that regulate the polarized delivery of LAT to the immune synapse rather than the lateral diffusion of the pre-existing plasma membrane pool of LAT [55]. The question of the mechanisms involved in the internalization of LAT is still unresolved. LAT, which has been shown to be present in cholesterol-rich domains of the plasma membrane [56], may be internalized in a clathrin-independent manner. Of note, in a recent review, Shafaq-Zadah et al. have pointed several links between the retrograde pathway and the mechanism of endocytosis that do not depend on the conventional clathrin machinery [57]. It would thus be important to characterize the endocytic pathways followed by LAT in resting and activated conditions as was recently done for the TCR [58]. Thus, the mechanisms of LAT trafficking from the plasma membrane to the IS, still need to be elucidated.

In summary, our work points to the Golgi apparatus as a major hub for the transport of LAT adaptor and describes how retrograde and anterograde pathways are organized and complementary, mechanistically and temporarily to coordinate the traffic of LAT-vesicles to the IS. It also helps in defining the finely-tuned mechanisms of transport of this adaptor during T-cell activation.

## Figures and Tables

**Figure 1 cells-10-00359-f001:**
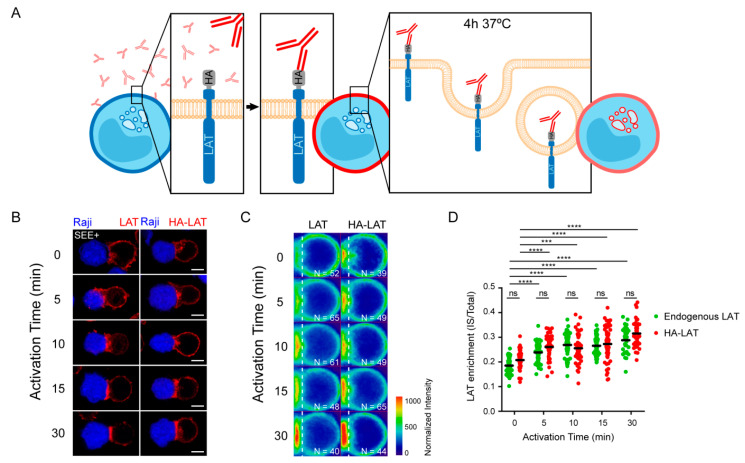
Total and endocytosed LAT are recruited to IS with similar kinetic. (**A**) Endocytosed LAT assay: Jurkat cells expressing a chimeric LAT protein tagged in its extracellular domain with HA (HA-LAT-Jurkat) were labeled at 4 °C for 30 min with a mouse anti-HA Ab, washed and then incubated at 37 °C for 4 h to allow LAT endocytosis. (**B**) Confocal images showing total (left panel) or endocytosed LAT (right panel) recruitment at the immune synapse (IS). Non-transduced Jurkat or HA-LAT-Jurkat cells after endocytosed LAT assay, were activated with SEE pulsed Raji B cells at different time points. Immunolabelings were performed using anti-mouse Ig (Alexa Fluor 568) to label the anti-HA Ab or anti-LAT to label total LAT. (**C**) Average cell representation and (**D**) quantification of the enrichment of total and endocytic LAT at the immune synapse (depicted by the dotted white line) in Jurkat T cells interacting with Raji B cells and activated with SEE for different time points. *n* = number of cells constituting the mean image. Horizontal lines represent the median. Scale bars = 5 μm. Two-way ANOVA *** *p* < 0.001, **** *p* < 0.0001, ns: nonsignificant. Data and images are from two independent experiments in (**B**–**D**).

**Figure 2 cells-10-00359-f002:**
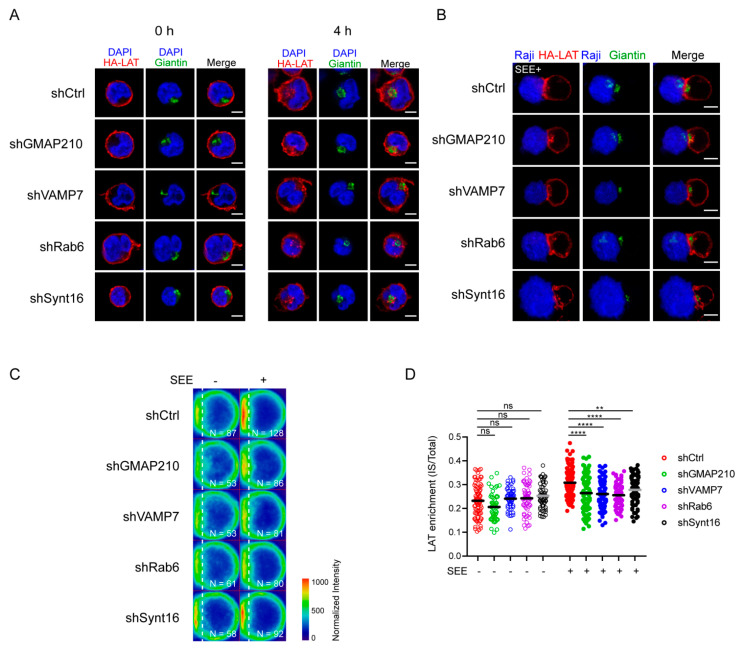
Retrograde and anterograde pathways regulate the recruitment of endocytic pool of LAT at the IS. (**A**) Confocal images of HA-LAT-Jurkat cells transduced with nontargeting control shRNA (shRNA Ctrl) or GMAP210/VAMP7/Rab6/Synt16-targeting shRNAs, fixed before (left panel, 0 h) or after endocytosed LAT assay (right panel, 4 h) and stained for endocytosed LAT (HA-LAT, red) and the Golgi apparatus marker Giantin (green). (**B**) Confocal images of conjugates between HA-LAT-Jurkat cells (after endocytosed LAT assay) expressing control (Ctrl) or GMAP210/VAMP7/Rab6/Synt16 specific shRNA, and SEE pulsed Raji cells (blue), labeled with anti-Giantin (green) and anti-mouse 568 (red). (**C**) Average cell representation and (**D**) quantification of the enrichment of endocytosed LAT at the immune synapse (depicted by the dotted white line) in control or GMAP210/VAMP7/Rab6/Synt16 silenced cells incubated with unpulsed (−, unactivated state) or SEE pulsed (+, immune synapse formation) Raji cells for 30 min. *n* = number of cells constituting the mean image. Horizontal lines represent the median. Images in (**A**,**B**) show the z-projection of summed slices from three stacks covering the Golgi apparatus in T cells. Scale bars = 5 μm. Two-way ANOVA ** *p* < 0.01, **** *p* < 0.0001, ns: nonsignificant. Data and images are from two independent experiments in (**A**) and from three independent experiments (**B**–**D**)

**Figure 3 cells-10-00359-f003:**
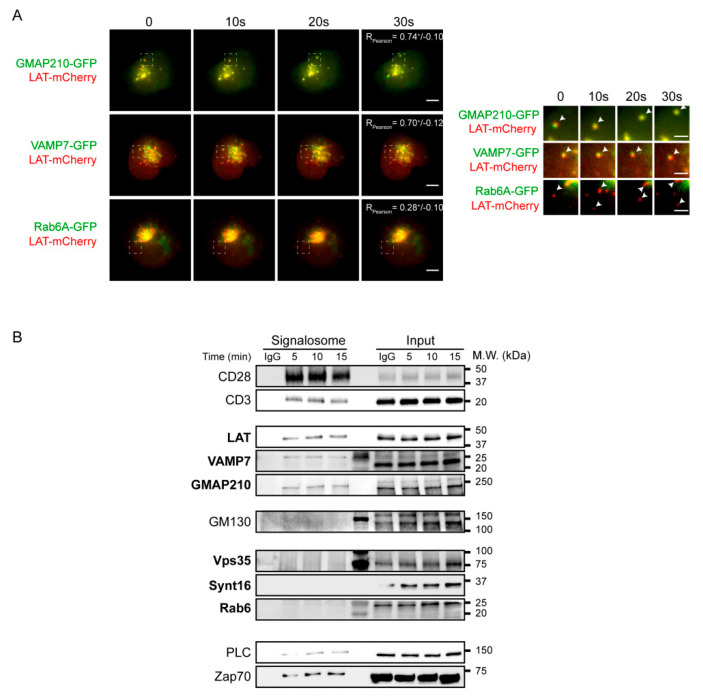
Anterograde, but not retrograde, pathway regulators are recruited together with LAT at the immune synapse. (**A**) Still images from live TIRFM imaging of Jurkat cells co-expressing LAT-mCherry and GMAP210-GFP, VAMP7-GFP or Rab6-GFP, seeded on coverslips coated with anti-CD3ε+anti-CD28 Abs. Pearson correlation coefficient is shown (Mean ± SD). Dashed square indicate the magnified regions. Arrow heads point out the appearance and displacement in the evanescent field of vesicles-containing simultaneous or individual LAT and GMAP210/VAMP7 or Rab6, respectively. Scale bars = 5 μm. Inset scale bars = 2 μm. (**B**) Immunoblot of signalosomes prepared from Jurkat cells activated for 5, 10 or 15 min with anti-CD3ε+anti-CD28 coated magnetic beads. IgG corresponds to Jurkat cells incubated with irrelevant mAb coated magnetic beads for 15 min. Proteins attached to the beads were purified by magnetic sorting after freezing and thawing the cells. Presence of the different proteins in the corresponding cell lysates (with detergent) are shown in “input” lanes. Data and images represent two independent experiments in (**A**,**B**).

**Figure 4 cells-10-00359-f004:**
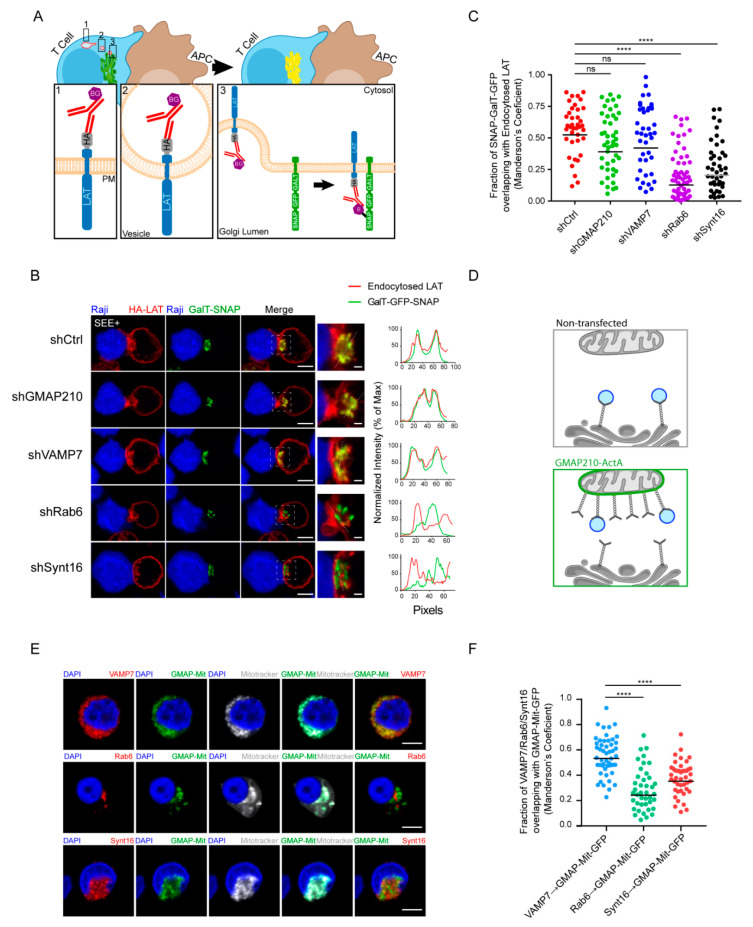
GMAP210 and VAMP7 do not regulate the retrograde pathway. (**A**) SNAP trap assay: Jurkat cells expressing GalT-GFP-SNAP and HA-LAT were stained at 4 °C with anti-HA Ab, washed, and incubated at 4 °C with BG-PEG9-NHS. After washing, cells were activated on slides for 30 min with Raji cells pulsed with SEE. Endocytosed membrane proteins, coupled with BG, covalently bound to the SNAP domain of the GalT-GFP-SNAP protein getting trapped in the Golgi. (**B**) Confocal images of SNAP trap assay in Jurkat cells expressing GalT-GFP-SNAP, HA-LAT and control (Ctrl) or GMAP210/VAMP7/Rab6/Synt16 specific shRNA. Labelings were performed using anti-mouse Ig (Alexa Fluor 568) to label the anti-HA Ab and anti-GFP to label the GalT-GFP-SNAP. Dashed square points to the inset localization containing the Golgi apparatus. Dashed line indicates where HA-LAT (red) and GalT-GFP-SNAP fluorescence signal intensity was measured for the plot. Images show the z-projection of summed slices from three stacks covering the Golgi apparatus. Scale bars = 5 μm. Inset scale bars = 1 μm. (**C**) Quantification of Mander’s overlapping coefficient of GalT-GFP-SNAP over HA-LAT. (**D**), Mitochondrial trapping assay: Jurkat cells expressing a chimeric GFP tagged GMAP210 protein with the C-terminal hydrophobic anchor of ActA can capture vesicles interacting with GMAP210 in the mitochondria. (**E**) Confocal images showing the localization of VAMP7, Rab6 or Synt16 (red) in Jurkat cells expressing a GFP-GMAP210-ActA chimera (GMAP-Mit, green), treated for 4 h with nocodazol (5 μg/mL) (nucleus in blue and mitochondria in gray). Scale bar 5 μm. (**F**) Quantification of Mander’s overlapping coefficient of GMAP-Mit with VAMP7, Rab6 or Synt16. Each dot represents one cell; horizontal lines represent the geomean. **** *p* < 0.0001, ns: nonsignificant (Kruskwal−Wallis test). Data and images represent one experiment in (**C**,**D**) and two independent experiments in (**E**,**F**).

**Figure 5 cells-10-00359-f005:**
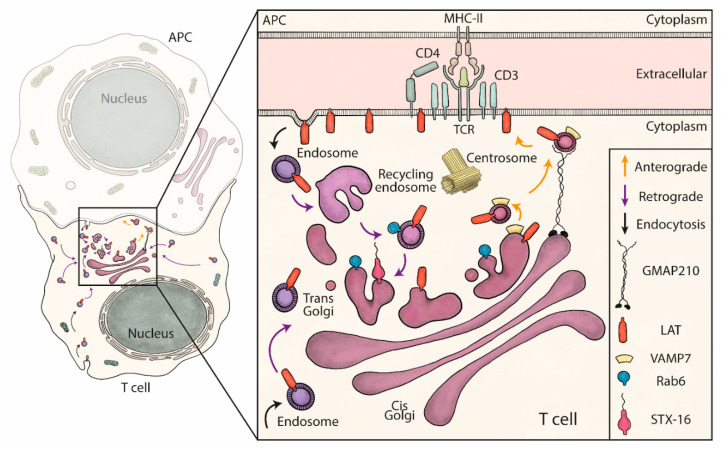
Schematic summary: retrograde and anterograde pathways coordinate the delivery of vesicles containing LAT to the immune synapse. LAT constitutively recycles from the plasma membrane to early/recycling endosomes. Vesicles from this compartment undergo a retrograde transport to the Golgi apparatus, which is increased upon TCR activation. Rab6 and Syntaxin-16 (inset), control specifically this retrograde transport. Once in the Golgi, the anterograde pathway delivers LAT to the IS. In this second step, LAT meets in the Golgi the vesicular SNARE VAMP7 and GMAP210 sorts, captures and brings the LAT/VAMP7 vesicles to the immune synapse.

## Data Availability

Not applicable.

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
