# Peer review of "Retrograde and Anterograde Transport of Lat-Vesicles during the Immunological Synapse Formation: Defining the Finely-Tuned Mechanism"

_cells, 2021, doi:10.3390/cells10020359_

Round 1

Reviewer 1 Report

The research topic is novel. The study shows an interesting potential mechanism of the recruitment of LAT to the IS. However, there is insufficient interpretation of the results on the potential of the findings in the paper.

In Figure 1C, only one shRNA is used in each group. Would you confirm that the phenotype is not due to off-target effects?

In Figure 1D, in shSynt16 group of Raji cells, the size of cells is bigger than the other groups, but in Figure 1C shSynt16 cells are smaller, is it affected the process of recruitment? Also, Giantin is presented in both Raji cell and Jurkat cells of shCtrl group, shRab6 group and shGMAP210 group, compared with other groups. Would you please explain this?

In Figure S1, it seems like there are more than two bands in figure A, but only two bands are well labeled. Also, there is no error bar in the shCtrl group. Would you explain the quantitative analysis of the result?

In Figure 3E, why the Mitotracker is co-localized with DAPI (Synt16 group), which means the mitochondria located in the nuclear? However, in Figure S2-B, it is located in the cytoplasm.

Reviewer 2 Report

The efficiency of T cell activation is controlled by the sub-cellular distributions of signalling intermediates. An important component of these distributions is the vesicular trafficking of the proximal adaptor protein LAT. Building on leading previous work from the Zucchetti/Hivroz groups, Saez et al comparatively investigate the role of four regulators of vesicular trafficking, Rab6, Syntaxin16, the Golgin GMAP210 and VAMP7, in LAT in retrograde transport from the plasma membrane to the Golgi and anterograde transport back to the plasma membrane. Using unique and powerful assays to investigate the individual trafficking steps in combination with shRNA-mediated knockdown of the four regulators of vesicular trafficking, the authors find that roles of Rab6 and Syntaxin16 are limited to retrograde transport, where as GMAP210 and VAMP7 control anterograde transport, thus establishing the existence of two distinct molecular machines in LAT trafficking. The comprehensive comparative nature of the experiments makes this an important contribution to the field. There are a few moderate suggestions for revision.

Fig. 1B establishes that the endocytosed pool of LAT behaves similar to all of LAT using representative images. A quantification similar to Fig. 1E/F would be helpful. The representative images suggest that LAT/HA-LAT accumulation at the interface is focussed on the interface centre at the 5 and 10 min time points but not later. While this is largely irrelevant to the conclusions of this manuscript, a quantification of central over all interface accumulation should be straightforward and would be interesting, in particular as central signalling accumulation, including that of LAT, has been controversially discussed in the literature for a long time. Any additional data are welcome.

In Fig. 1E/F it would be helpful to provide the statistical significance in the comparison of APCs with and without superantigen for the control and each of the four knockdown conditions. By inspection of the figure, LAT interface enrichment may remain inducible by superantigen for some of the conditions. If that turns out to be correct, differences in the antigen-dependence of interface LAT recruitment upon knockdown of the vesicular regulators should be discussed.

The authors discuss carefully how LAT trafficking involves routing through the Golgi and should be commended for the summary figure 4. This is important, as much vesicular trafficking of plasma membrane proteins, including that of the TCR, is thought to involve the direct plasma reinsertion from recycling endosomes, thus bypassing the Golgi. Saez et al should discuss more extensively evidence that this more direct recycling pathway does not play a major role for LAT, in other words how much LAT recycling uses the Golgi versus recycling endosome route.

As a very minor suggestion, it may be worthwhile adding the Pearson’s correlation coefficients to Fig. 2A.

Round 2

Reviewer 2 Report

I would like to thank the authors for the careful revision of the manuscript.